# Conflict-Induced Shocks and Household Food Security in Nigeria

**Opeyemi Olanrewaju and Bedru B. Balana \***

International Food Policy Research Institute (IFPRI), Abuja 901101, Nigeria
* Correspondence: b.balana@cgiar.org

**Abstract:** Conflicts such as the Boko Haram insurgency, herder–farmer conflicts, and armed banditry attacks are major concerns affecting the livelihoods and food security of households in Nigeria. In this paper, firstly, we reviewed and synthesized the nature, spatial extent, and implications of conflicts on food security in Nigeria. Secondly, using survey data and econometric models, we examined the effects of conflict-induced shocks, such as forced migration and fatality on household food security indicators. Our review shows that the underlying causes for the majority of violent conflicts in Nigeria are linked to competition for productive resources, economic inequality, and ethnoreligious tensions. Review results also indicate spatial variations in the nature and severity of violent conflicts in Nigeria. While the Boko Haram insurgency is prominent in the North-East, the North-Central is mainly exposed to herder–farmer conflicts, and there is a high prevalence of communal conflicts in the South-South region of the country. In terms of gender dimensions, women are more vulnerable to conflicts and shoulder more social and economic burdens than men. From our empirical analysis, we found that conflict-induced shocks such as forced migration, fatality, abduction, and injury significantly exacerbate the severity of food insecurity and deteriorate the dietary diversity of households. Conflicts also affect agricultural investment decisions with a negative consequence on future agricultural productivity and food security. Based on the findings, the key policy suggestions include the need for tailored interventions to resolve state or region-specific conflicts, policy interventions on property/land rights and livestock management systems to address herder–farmer conflicts, and targeted investments in building the resilience capacity of households.

**Keywords:** conflict-induced shocks; dietary diversity; food security; forced migration; North-East Nigeria

## 1. Introduction

Conflicts and general security threats, including herder–farmer conflicts, the Boko Haram insurgency, armed banditry attacks, and kidnappings are major security concerns in Nigeria affecting the livelihoods of households, agricultural investment, production activities, productivity, and food security. Recent studies show that conflicts and terrorist attacks reduced the area cultivated, agricultural output and productivity, and investments [1–3]. Conflicts also reduced farmers' cattle holdings by increasing cattle thefts and losses and reducing purchased cattle [4]. For instance, the herder–farmer conflict resulted in intense competition to land and led to clashes among herders and farmers in many parts of Nigeria [4,5]. Herder–farmer conflicts appear to be damning, deeply rooted, and widespread in Nigeria. This is because the livelihoods of over 70 percent of the Nigerian population depend on agriculture and thus there are growing conflicts over access to resources (land and water) [6]. When such conflicts are not well-managed, they degenerate into violence and destructive social clashes that disrupt economic activities, deteriorate livelihoods, and worsen food insecurity [7].

Aside from the herder–farmer conflicts, the Boko Haram insurgency is another major source of insecurity in Nigeria. According to the Armed Conflict Location and Event Data Project (ACLED) [8] through the end of 2016, northeastern Nigeria, which is the hotspot for Boko Haram insurgents, recorded over 30,000 deaths. A recent study on a

sample of 1500 internally displaced persons (IDPs) in Borno state in northeast Nigeria revealed that 85 percent of the surveyed IDPs identified the Boko Haram insurgency to have contributed to their food insecurity [9]. The insurgency contributes to food insecurity through various pathways in food production, such as by deterring farmers to access their farms and delaying critical farm operations [10,11]. In addition to the above two major types of conflict, i.e., the herder–farmer conflict and Boko Haram insurgency, armed banditry attacks and kidnappings have brought a new and evolving dimension to the issue of general insecurity and conflicts in Nigeria. The frequency and spatial scale of armed banditry attacks have been increasing over the past few years and hence they pose devastating effects on the livelihoods and food security of households in Nigeria [12].

Based on the existing knowledge that violent conflict simultaneously impacts livelihoods and food security [1–3], this study aims to add empirical evidence on the relationship between conflict-induced shocks and the severity of household food (in)security, using Nigeria as a case study country. Nigeria is an interesting country for such a study because of the protracted conflicts, such as herder–farmer conflicts and the Boko Haram insurgency, that caused significant economic damage, losses of human lives, and food security of households. While previous studies [13] analyze how overall conflict circumstances affect efforts to combat food insecurity, the linkages between specific conflict-induced shocks and household food security have not been examined, especially within the context of violent conflict settings. Furthermore, evidence on the relationship between household food (in)security and the direct effects of conflict-induced shocks such as forced migration, fatality, loss of property, and injury, among others, as well as the role these factors play in different routes to food insecurity has not been sufficiently examined. Therefore, the purpose of this study is to examine how conflict-induced shocks mediate different pathways to food insecurity in conflict-affected areas. Such an understanding is highly critical for developing evidence-based policy responses to mitigate its negative impacts and design long-term recovery strategies.

The remaining parts of the paper are organized as follows. Section 2 presents the conceptual background that underpins the drivers and impacts of conflicts in Nigeria. A brief overview of the relationship between conflict and food security is presented in Section 3. The methodological part of the paper including the data (the 2018 FADAM-III data and IFPRI's 2021 phone survey data during COVID-19 in Nigeria) and econometric estimation strategy are presented in Section 4. Section 5 presents the empirical findings, and the Section 6 concludes the paper with some policy suggestions.

## 2. Conceptualizing Conflicts in Nigeria

### 2.1. Drivers and Impacts

The nature in which a conflict expresses itself depends on the drivers and processes through which it originates and the groups involved in it. Figure 1 exhibits the conceptual framework of the causes, drivers, and impacts of conflicts. Researchers present the drivers and conflict–food security nexus in different ways. Ningxin [14], for example, emphasizes historical, ideological/cultural, ethnic, and religious factors as the key drivers of conflicts. On the other hand, Abdul [15] emphasizes competition over resources, inadequate information, psychological needs, and values as the key factors driving conflicts. Some conflicts could be between the same resource user groups, while others could be between different user groups. Prominent among the same resource user groups is the one between neighboring communities sharing a common grazing land. In the case of different user groups, the most common is between farmers and herders over land [16]. Studies show that the struggle over natural resources such as land is a primary source of violent conflicts among communities in Nigeria [17–19]. The conflicting parties often see each other as trespassers on their lands. For example, in the case of the herder–farmer conflict, the herders believe they have the right to use the land and, therefore, dismiss the other party (usually farmers) as mere trespassers on their land. On the other hand, many farmers consider herders as strangers who are occupying their land [18].

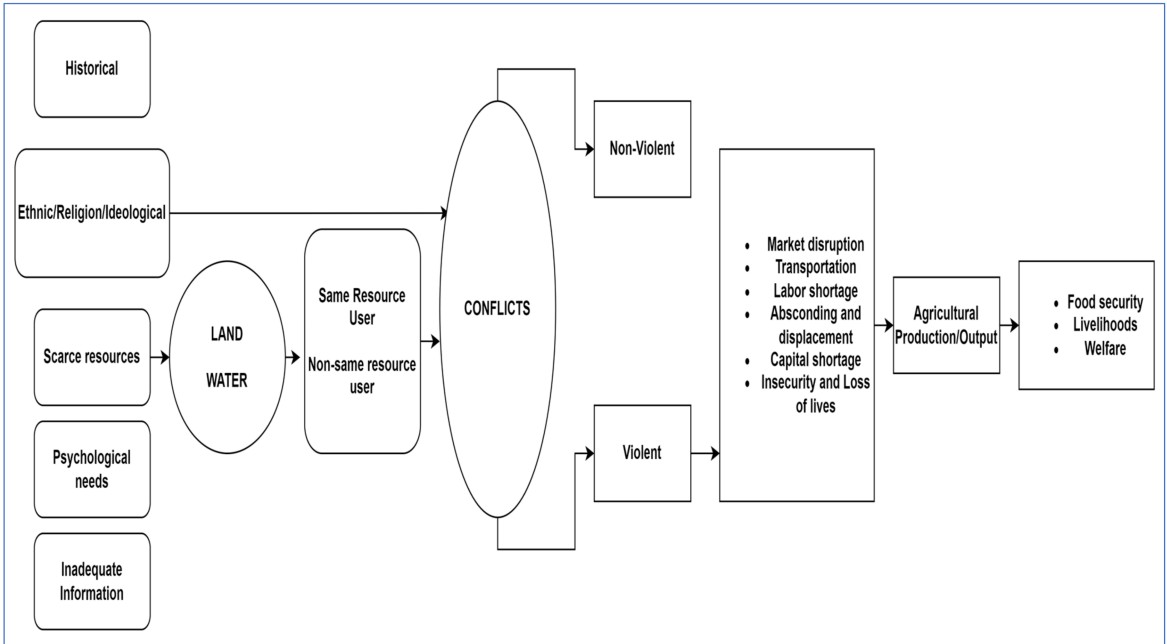

**Figure 1.** Conceptual framework of the drivers and impacts of conflict in Nigeria.

The herder–farmer conflict is further fueled by growing suspicion between the two parties. The pastoralist herders travel hundreds of miles from the northern part of the country with their cattle in search of grazing land. The herders believe that farmers often steal their herds and as a result arm themselves with weapons to protect their territory and livestock. On the other hand, in anticipation of attacks from herders, the farmers attempt to deter the herders [20]. Historically, the competition for land led to tensions and remains unabated over the years between both parties in Nigeria.

Manipulation of information has also been found to have contributed to conflicts in societies. Often information can be distorted and, therefore, be either manipulative or constructive. Given the critical role that information plays in society, when it is tampered with, conflicts tend to arise. Information is often tampered with when people are being fed lies and wrong information. This distorted information shape opinions and decisions and could influence the nature, scope, and intensity of conflicts. Another factors fueling conflicts are ethnic and religious differences, which could further be manipulated by ethnic purveyors and politicians to leverage as tools for their economic and political goals [21]. As depicted in Figure 1, violent conflicts could lead to the destruction of livelihoods, severe food insecurity, and welfare loss.

The conflicts in Nigeria exhibit features of geographical dimensions too. While most parts of Nigeria experience conflicts in one or the other forms, the intensity and the preponderant exposure to conflict vary across the various regions of the country. Of the six geopolitical zones of Nigeria, the North-East (NE), North-Central (NC), and South-South (SS) zones appear to be the most preponderant areas of conflict in the country. NE Nigeria has witnessed several forms of conflicts of which the majority are traced to terrorist attacks [22]. The NC region is mainly plagued with the farmer–herder crisis [12,23,24]. Some pockets of the NC region also experience attacks from Boko Haram insurgents. The SS region is renowned for being Nigeria's source of oil wealth and is widely regarded as the fulcrum of Nigeria's oil economy. The majority of the conflicts in the SS region are attributed to communal violence, criminals, and non-organized attacks by individuals [22]. Many of the reasons for the violence in the SS region can be traced to the struggle among local communities related to control over oil revenues derived from this resource-rich territory. Consequently, nearly one-third of conflict-affected households in the SS region have been displaced or experienced forced migration [22].

Conflicts pose differential impacts on women and children. Studies affirm that regardless of the types and severity of conflicts, women and children are more vulnerable in the face of conflicts than men [25,26]. The pre-existing unequal access to assets and resources between women and men contributes to the high vulnerability of women in conflict-exposed communities [27]. Particularly, women bear the brunt of conflicts and insecurity as they are forced to take more economic responsibilities in the demise of their husbands or working male household members who become casualties of conflicts. The death of household members of working age means that the households become female-headed, with limited opportunities to access resources for their livelihood due to sociocultural barriers to women's access to resources [28]. Women's vulnerability could sustain for a longer period if their means of livelihood are significantly affected by the conflicts, and they lack the capacity to rebuild their livelihoods and well-being [29].

### 2.2. Hypotheses

Based on our conceptual depiction and understanding of existing research evidence, we hypothesize two propositions to guide our investigation on the key drivers of conflict on the one hand, and on the linkages between conflicts and household food security on the other.

**Hypothesis 1.** *Several factors such as ethnoreligious differences, power struggles or political motivations, and growing economic inequality could trigger conflicts in Nigeria. However, by synthesizing the existing evidence, we hypothesize that the struggle over natural resources such as land is the primary source of violent conflict among communities in Nigeria. In other words, competition for productive resources is the key driver for the majority of violent conflicts in Nigeria.*

**Hypothesis 2.** *We hypothesize that conflict-induced hocks such as forced migration, death of household members, loss of property, injury, and the abduction of household members due to conflict increase the likelihood of households falling into a more severe food insecurity situation.*

## 3. Overview of the Effects of Conflicts on Food Security

Conflicts have dire consequences on the livelihoods and food security of vulnerable households. They drastically reduce agricultural production and as a result have grave implications for the food security of households, particularly the poor and vulnerable ones [30,31]. Beyond limiting the production of food, conflicts also have the potential to hamper food availability and supply [32]. Conflicts reinforce the vicious circle of extreme poverty and exacerbate the food insecurity of vulnerable social groups (i.e., women and children) and most marginalized groups [33]. The FAO and others [34] affirm that there is a strong positive correlation between violent conflicts and food insecurity, as all nineteen countries classified by the FAO as 'protracted crisis' conditions in 2017 were engaged in violent conflicts. According to the Uppsala conflict data program [35], countries such as South Sudan, Somalia, and Yemen that are exposed to significant violent conflicts also experience a high risk of famine. Conflict-ridden regions are home to over 60 percent of the world's hungry population [36]. Furthermore, low- and middle-income countries affected by conflicts have an average prevalence of undernourishment between 1.4 and 4.4 percent higher than conflict-free nations in the same income bracket [36].

Generally, depending on the intensity and scope, violent conflicts take diverse forms and have varying implications for food security. At the national scale, they have the tendency to affect all four aspects of food security—availability, access, utilization, and stability—whereas at the local level, such as in northeastern Nigeria, they may have a relatively larger impact on the availability and access to food than food utilization [37]. Food security is typically shown to be impacted by communal conflicts by lowering agricultural output and household income. Additionally, it tends to restrict people's access to the food supply chain and availability [38]. The relationships between food insecurity and violent conflict are also characterized by a high degree of intricacy and contextualization, often



coinciding with multi-layered crises. For instance, violent conflicts are also a major cause of forced displacement, in addition to food shortages and starvation [39], which weakens food security in both the communities of origin (where labor may be in short supply and rural markets collapse) and the host communities (which may face pre-existing strong pressure on limited arable land). For example, the disruption and spiking food prices caused by the Russia–Ukraine conflict negatively affected the food security of countries such as Nigeria, which depends largely on imported grain (e.g., only 1 percent of the wheat consumed in Nigeria comes from domestic production). The war in Syria has forced more than 6 million people to flee their homes, which led to the refugee and food security crisis [40]. Globally, there were 11.6 million refugees in protracted conflicts in 2016 and 13.4 million in 2017 [41]. Of these, 6.5 million have been displaced for more than a decade [42].

Conflicts can lead to the destruction of farmland, livestock damage, crop theft, destabilize food markets [43], limit household diet diversity [44], and impair food security [39,45]. Conflicts indirectly affect food insecurity through several mechanisms, including interfering with agricultural production [46] and influencing farmers' investment choices [47]. Furthermore, households afflicted by violence frequently also experience non-conflict shocks, which undermines the link between conflict and food insecurity, such as economic instability [48]. To mitigate the dire impacts of conflicts on food security, households may adopt various negative coping strategies, such as eating less nutritious food with more calories or having a less varied diet. For instance, using cross-sectional data, Dabalen and Paul [49] evaluated the impact of conflict on dietary diversity in Côte d'Ivoire and found that households with individuals who live in the most severely affected conflict zones have decreased dietary diversity. Another study [13] examined the effects of the Boko Haram insurgency on food insecurity conditions using panel survey data from Nigeria and found that the insurgency decreased the availability of production input and income, increased the number of days households had to rely on less preferred foods, restricted the variety of foods eaten and the portion sizes consumed, and decreased dietary diversity, as measured by the food consumption score. Such negative coping strategies of households adopted in conflict-affected settings have been established in various other studies [28,39,46,47].

Conflict is one of the major drivers of displacement and forced migration, ultimately leading to severe food insecurity [50]. The World Food Program report [50] shows a strong connection between conflict, forced migration, and food insecurity. According to the Report, the number of refugees per 1000 people increased by 0.4 percent in 2017 for every year of conflict and by 1.9 percent for every percentage rise in food insecurity [50]. According to the FAO [51], when conflicts worsen food and nutrition security, there is a greater chance that the conflict would intensify and last longer.

Thus, the growing difficulties of achieving food security in violent conflict-affected settings described in the preceding paragraphs suggest a positive relationship between conflicts and an increase in the severity of food insecurity [52–54].

## 4. Materials and Methods

In this section, we present the econometric methods implemented for the analysis of the effects of conflict-induced shocks on food security measures using the data from the 2018 Fadama-III survey and phone survey data collected by IFPRI in 2021.

### 4.1. Data and Measurement of Variables

This paper employs data from the World Bank-funded project Fadama III, Phase II, which was implemented in the conflict-affected North-East Nigeria states of Borno, Yobe, Adamawa, Taraba, Bauchi, and Gombe. The World Bank Fadama III Additional-Financing dataset was collected in 2018 by IFPRI as part of the World Bank-funded project (Fadama III-Additional Financing (AF II) Phase II) that was implemented in North-East Nigeria (NE). The project was supporting the recovery of the agricultural sector in the NE and responding to urgent food and livelihood needs of farming households affected by conflicts. The region suffers huge economic damages and losses of human life owing to persistent

violent conflicts. The North-East states are renowned for their large agricultural potential. However, the region has suffered from prolonged conflicts that led the region to significant setbacks in terms of economic and social development. To assess the effects of conflicts on household food security, we extract variables from the Fadama III Phase-II dataset. Table 1 provides descriptions and summary statistics of variables used in the empirical analysis. While the survey sampled a total of 1800 households, we used a sample of 1658 households with complete data across the six states of the North-East region. The survey data span information across violent conflicts, migration, socioeconomic conditions, credit access, and humanitarian support received, among others (Table 1). In addition to the Fadama-III data, we also used the data IFPRI collected in 2021 via a phone interview from a sample (n = 1031) of households in four states of Nigeria during the COVID-19 pandemic.

We employed the eight standard experience-based food insecurity experience indicators for measuring food security [55]. Indicators such as these have been widely employed in the investigation of food insecurity [56–58]. Out of the eight standard experience-based food insecurity questions, we concentrated on three indicators that reveal households' most severe food insecurity experiences over a four-week period prior to the survey date.

1. 'Was there a time when your household ran out of food because of a lack of money or other resources? (yes/no)'
2. 'Was there a time when you or others in your household were hungry but did not eat because of a lack of money or other resources? (yes/no)'
3. 'Was there a time when you or others in your household went without eating for a whole day because of a lack of money or other resources? (yes/no)'

For the dietary diversity indicator, following Swindale et al. [59], we created a household dietary diversity score (HDDS) utilizing the "yes/no" responses to the 12 food groups that were consumed by a household over a specified reference period. The HDDS was created by summing horizontally a binary response, "yes = 1" if the household ingested any food from the particular food group during the reference period, and "no = 0" otherwise. As a result, the HDDS has a minimum value of zero and a maximum value of twelve.

**Table 1.** Descriptions/measurements of the variables used in the models (n = 1658).

| Variables | Descriptions | Mean [1] |
|---|---|---|
| *Dependent variables (food (in)security measures)* | | |
| Household dietary diversity score (HDDS) | Food consumption scores across the 12 food groups (continuous) | 6.671 |
| Run out of food | Households 'run out of food' within the last 4 weeks prior to the survey (yes = 1, no = 0) | 0.343 |
| Hungry but did not eat | Households that any member 'went to sleep at night hungry' in the last 4 weeks prior to the survey (yes = 1, no = 0) | 0.327 |
| Without eating for a whole day | Households that any of its household members 'went a whole day and night without food' in the last 4 weeks prior to the survey (yes = 1, no = 0) | 0.218 |

**Table 1.** *Cont.*

| Variables | Descriptions | Mean [1] |
|---|---|---|
| *Explanatory variables* | | |
| Displaced | Households that any member of its household was displaced because of violent conflicts [2] (yes = 1, no = 0) | 0.297 |
| Abducted | Households that any member of its household was abducted owing to violent conflicts (yes = 1, no = 0) | 0.0718 |
| Trauma | Households who reported any member of its household abducted owing to violent conflicts (yes = 1, otherwise = 0) | 0.2159 |
| Fatality | Households where any member of its household was killed owing to violent conflicts (yes = 1, otherwise = 0) | 0.1470 |
| Loss of property | Households who lost their properties owing to violent conflicts (yes = 1, no = 0) | 0.522 |
| Injured | Households that any member of its household was injured owing to violent conflicts (yes = 1, otherwise = 0) | 0.242 |
| Migrated because of insecurity | Households who migrated owing to violent conflicts (yes = 1, no = 0). | 0.432 |
| Received humanitarian assistance | Households who received any form of humanitarian assistance (yes = 1, no = 0) | 0.481 |
| Received credit | Household who received credit (yes = 1, no = 0) | 0.0759 |
| Access to extension services | Household who accessed extension services (yes = 1, no = 0) | 0.0633 |
| Loss of market infrastructure due to conflicts | Households who reported destruction to their community market owing to conflicts (yes = 1, no = 0) | 0.276 |
| Own agricultural processing equipment | Households who own any agricultural processing equipment (yes = 1, no = 0) | 0.033 |
| Age of household head | Age of household head in years (in years) | 48.039 |
| Education level of household | If the household head is educated up to secondary school level (yes = 1, no = 0) | |
| Access to market information | Households who have access to market information (yes = 1, no = 0) | 0.646 |
| Household involvement in non-farm activities | Households that are involved in non-farm activities for livelihood (yes = 1, no = 0) | 0.113 |

Source: Authors' compilation from the Fadama-III survey (2018). Note: [1] means of dummy variables are percentages of 'yes' responses. [2] Violent conflicts considered here include Boko Haram, armed conflicts, and tribal conflicts.

### 4.2. Empirical Models

#### 4.2.1. Probit Model

For the three binary experience-based food insecurity indicators ('run out of food' 'hungry but did not eat', and 'without eating whole day'), we used a basic binary outcome probit model in Equation (1) to predict the likelihood of a household experiencing the severe food insecurity situations [60,61]. Suppose that the outcome variable, i.e., the probability that the household experience a given food insecurity indicator, denoted by $y$, takes one of two values:

$$y = \begin{cases} 1 & \textit{if a household experienced food insecurity} \\ 0 & \textit{if a household did not experience food insecurity} \end{cases}$$

Following Greene [62] and assuming a normal distribution of the error term in the mode and given a vector of explanatory variables denoted by $x$, the probability that $y = 1$, i.e., the conditional probit probability ($P$) takes the form in Equation (1).

$$P(y = 1/x) = F(x'\beta) + u \tag{1}$$

where $y = 1$ if a household experienced 'run out of food', 'hungry but did not eat', or 'without eating whole day', and zero otherwise, $F(.)$ is a cumulative parametric function, $\beta$ stands for coefficients, and $u$ stands for a normally distributed random error term. Following Greene [62], we estimate the marginal effects of explanatory variables as the effect of a unit change of the specific variable $x_i$ on the conditional probability $P(Y = 1|x_i)$, given that all other variables are constant, as in Equation (2) (see [60]).

$$\partial P(y = 1|x_i)\partial x_i = \partial E(y|x_i)/\partial x_i = \varphi(x'\beta)\beta_i \tag{2}$$

### 4.2.2. Negative Binomial (NB) Model

The HDDS exhibits the features of count data. Thus, we employ the negative binomial (NB) model following [60]. The probability of the dependent variable Y takes the value of $y$, i.e., $\Pr(Y = y)$ can be specified using the Poisson model, as in (Equation (3)):

$$\Pr(Y = y) = \frac{e^{-u}\mu^{-y}}{y!} \tag{3}$$

where $Y$ is the dependent variable that takes the value of $y$, i.e., $\Pr(Y = y)$; $\mu > 0$ with exponential mean parametrization as $\mu = exp(x_i'\beta)$; $y = 1, 2, \ldots, 12$ representing the HDDS values, and $x'$ is the set of independent covariates, as specified in Table 1. To relax this restrictive property of the Poisson model (the equality of mean and variance (i.e., $E(Y) = Var(Y) = \mu$)), we adopt the less restrictive quadratic variance negative binomial model that accommodates overdispersion [60] using the '*nbreg*' Stata command.

## 5. Results and Discussion

### 5.1. Descriptive Summary of Key Food Security Indicators

Table 2 reports summary statistics on mean differences for the four food security indicator variables used in the econometric models. The results show that the average HDDS of households who have migrated due to violent conflicts within the last ten years is 5.88 against 7.32 for those who did not migrate, and the difference is statistically significant at a 1 percent level.

**Table 2.** Mean differences in food security indicators (by conflict-induced migration status).

| Variables (n = 1658) | Pooled Mean | HH Migrated Due to Conflicts (Mean1) | HH Did Not Migrate (Mean2) | Difference (Mean1–Mean2) |
|---|---|---|---|---|
| Household dietary diversity score (HDDS) | 6.671 | 5.818 | 7.319 | 1.501 *** |
| Run out of food | 0.343 | 0.444 | 0.266 | −0.178 *** |
| Hungry but did not eat | 0.218 | 0.312 | 0.147 | −0.166 *** |
| Without eating whole day | 0.327 | 0.425 | 0.252 | −0.174 *** |

Source: Authors' compilation from the Fadama-III survey (2018). Note: HH = household. *** $p < 0.01$,

Statistically significant differences are also observed in all three experienced-based measures of food insecurity. Our results show that about 26.6 percent of households who did not migrate due to conflicts reported their household experienced '*run out of food*' in the last 4 weeks prior to the survey compared to 44 percent of households who migrated due to conflicts. In terms of the '*hungry but did not eat*' indicator, we also found a statistically

significant difference between the two household groups, 31 percent for migrated vs. 15 percent for households that did not migrate.

Comparing the two household groups in the most severe food insecurity indicator, '*without eating the whole day*', about 43 percent of households that experienced conflict-induced migration have experienced that members of their households 'went the whole day without eating' anything compared to 25 percent for households that did not face conflict-induced migration.

### 5.2. The Effects of Conflicts on Food Security Amid COVID-19

Climate change-related shocks and the COVID-19 crisis might have likely exacerbated the incidence of conflicts and subsequently affected the livelihoods and food security of households in Nigeria. Based on the responses to conflict-related questions in the phone interview conducted in July 2021 [63] in the four Nigerian states surveyed (Kebbi, Benue, Delta, and Ebonyi), on average nearly 50 percent of survey households experienced insecurity threats in the 12 months prior to the interview. Comparable results to ours in the northern states of Nigeria were reported by [64]. It should be noted, however, that the conflicts and insecurity in northern Nigeria have existed for over a decade before COVID-19; thus, we are cautious not to directly associate the rise in conflicts/insecurity threats with the pandemic. However, 73 percent of survey respondents indicated that the insecurity threats had increased over the last 12 months compared to the situation the year before COVID-19. As shown in Table 3, the agricultural activities of over one-third of the households surveyed were extremely or moderately severely affected by conflicts/insecurity. These reduce uses of yield-enhancing agricultural inputs leading to low agricultural output and could lead to increased severity of food insecurity.

**Table 3.** Effects of insecurity threats on agricultural activities.

| Questions: How Severely Has the Presence of Insecurity Threats Affected Your Household's: [ . . . . . . .] | Respondent's Subjective Assessment of Severity of Conflicts/Insecurity on Major Agricultural Activities and Markets (%) | | | | |
|---|---|---|---|---|---|
| | Extremely Severe [1] | Moderately Severe [2] | [1] + [2] | Slightly Severe [3] | Not at All [4] |
| 1. . . . access to *agricultural input markets*? | 18.33 | 17.26 | 36 | 20.83 | 43.57 |
| 2. . . . access to *market to sell agricultural produce*? | 16.79 | 16.31 | 33 | 21.07 | 45.83 |
| 3. . . . . *normal farm operations* (planting, ploughing, weeding, harvesting)? | 19.17 | 16.07 | 35 | 21.43 | 43.33 |
| 4. . . . . *farm investments* (e.g., expand cultivated area; more livestock)? | 18.93 | 15.12 | 34 | 21.10 | 44.76 |

Source: Authors' compilation from the phone survey data (July 2021).

### 5.3. Conflicts and Household Dietary Diversity Scores (HDDS)

Table 4 reports the estimates from the negative binomial (NB) model along with the marginal effects of the covariates. The regression coefficients, as reported in Table 3, are statistically significant at the 1 percent level (Wald Chi2 (16) test statistic, $p = 0.000$). Thus, the overall fit of the model is good. Eight of the sixteen regressors in the NB model are statistically significant at the 1, 5, or 10 percent levels. The four key violent conflict-induced factors we considered in the HDDS estimation include displacement, abduction, and loss of property owing to conflicts. We found that the HDDS as a household's food security measure is highly susceptible to conflicts and conflict-induced migration. A unit decrease in conflict-induced migration is more likely to increase household dietary diversity by about 12 percent. A 2017 World Food Program report similarly noted that the greatest refugee outflows are from countries that are experiencing armed conflicts and food insecurity [50].

George [13] also noted that violent conflicts trigger forced migration and displacement of people, and as a result present a potential channel for disrupting household welfare.

**Table 4.** The effects of conflict-induced migration on household dietary diversity score (HDDS).

| Variables | Coefficients | Robust Std. Error (Coef.) | Marginal Effects [†] | Std. Error (Marginal Effects) |
|---|---|---|---|---|
| Migrated due to conflicts | −0.1888 *** | 0.024 | −1.22 *** | 0.153 |
| Displaced | 0.0338 | 0.024 | 0.223 | 0.161 |
| Abducted | −0.128 *** | 0.044 | −0.796 *** | 0.258 |
| Loss of property | −0.110 *** | 0.028 | −0.722 *** | 0.184 |
| Received humanitarian assistance | −0.003 | 0.021 | −0.025 | 0.138 |
| Received credit | −0.0173 | 0.033 | −0.112 | 0.215 |
| Access to extension | −0.084 | 0.054 | −0.531 | 0.332 |
| Market infrastructure loss due to conflicts | 0.248 *** | 0.023 | 1.726 *** | 0.169 |
| Own agricultural processing equipment | −0.119 * | 0.066 | −0.740 * | 0.386 |
| Age of household head | 0.002 ** | 0.001 | 0.0133 ** | 0.006 |
| Education level of HH | −0.001 | 0.022 | −0.006 | 0.145 |
| Access to market information | 0.046 * | 0.024 | 0.299 * | 0.156 |
| HH involvement in non-farm activities | −0.0544 * | 0.0308 * | −0.348 | 0.1939 |

Source: Authors' compilation from the Fadama-III survey (2018). Note: HH = household. *** $p < 0.01$, ** $p < 0.05$, * $p < 0.1$. Note: [†] marginal effects (dy/dx) are evaluated at the sample values and then averaged.

Regarding other direct impacts of conflicts considered in the study, the abduction of household members negatively and significantly affects household food security. One plausible pathway for this could be when abducted household members include working household members that contribute significantly to household livelihoods. Similarly, households that have suffered property loss owing to security threats are also found to be susceptible to food insecurity, as our estimates show a significant negative relationship with household dietary diversity scores.

In addition to conflict-induced factors, we also considered several covariates in the estimation (Table 4). For instance, access to market information increased with household food security (at the 1 percent level) with an estimated marginal effect of 30 percent, underscoring the important role market information plays in improving household security, especially in times of violent conflicts when neighborhood market may be susceptible to a vicious cycle of violent attacks. Our finding is consistent with recent findings that demonstrated that households which are located closer to market centers are likely to be more food secure [65]. On the hand, our results show that households that received humanitarian support were not statistically different from those that did not receive support, implying that humanitarian assistance may be not sufficient to lift households out of food insecurity situations. This finding is consistent with Balana et al. [58], who showed that the safety net interventions during the COVID-19 pandemic in Nigeria did not provide statistically significant effects in improving household food security outcomes.

*5.4. Conflicts and Household Food Insecurity Experiences*

Results from the probit model for the three experience-based food insecurity indicators are presented in Table 5. Here, our dependent variables of interest were the households' 'yes/no' responses to the three food insecurity experience questions, as described in Section 2: (1) whether there has been a time the household '*ran out of food*' because of a lack of money or other resources (yes/no); (2) whether there has been a time any household member was '*hungry but did not eat*' because of a lack of money or other resources (yes/no);

and (3) whether there has been a time any household member *'went without eating for a whole day'* because of a lack of money or other resources (yes/no).

**Table 5.** Probit model results for food insecurity indicators.

| Variables | Ran Out of Food | | Without Eating for the Whole Day | | Hungry but Did Not Eat | |
|---|---|---|---|---|---|---|
| | Coefficient | Marginal Effects [†] | Coefficient | Marginal Effects [†] | Coefficient | Marginal Effects [†] |
| Migrated due to conflicts | 0.349 *** | 0.1264 *** | 0.3916 *** | 0.1069 *** | 0.3358 *** | 0.118 *** |
| | (0.071) | (0.0255) | (0.0772) | (0.0210) | (0.0715) | (0.025) |
| Displaced | −0.049 | −0.0179 | −0.2889 *** | −0.0788 *** | 0.06281 | 0.022 |
| | (0.084) | (0.0306) | (0.0912) | (0.0250) | (0.0843) | (0.029) |
| Abducted | −0.018 | −0.0068 | −0.0577 | −0.0157 | 0.02505 | 0.008 |
| | (0.134) | (0.0487) | (0.1403) | (0.03830) | (0.1323) | (0.046) |
| Trauma | 0.253 *** | 0.0919 *** | 0.1259 | 0.0344 | 0.2611 *** | 0.0924 *** |
| | (0.093) | (0.0339) | (0.1020) | (0.0279) | (0.0936) | (0.033) |
| Fatality | 0.619 *** | 0.2243 *** | 0.3751 *** | 0.1024 *** | 0.5619 *** | 0.199 *** |
| | (0.104) | (0.0376) | (0.1073) | (0.0293) | (0.1040) | (0.036) |
| Loss of property | −0.114 | −0.0413 | 0.21625 ** | 0.0590 ** | −0.0516 | 0.033 |
| | (0.083) | (0.0303) | (0.0916) | (0.0250) | (0.0841) | (0.029) |
| Injured | 0.195 ** | 0.0708 ** | 0.2821 *** | 0.0770 *** | 0.1064 | 0.037 |
| | (0.090) | (0.0327) | (0.0968) | (0.0263) | (0.0901) | (0.031) |
| Received assistance | −0.339 *** | −0.1230 *** | −0.1961 *** | −0.0535 *** | −0.3186 *** | −0.1128 *** |
| | (0.069) | (0.0249) | (0.0753) | (0.0206) | (0.0697) | (0.024) |
| Received credit | 0.227 * | 0.0825 * | 0.1469 | 0.0401 | 0.2419 * | 0.085 * |
| | (0.125) | (0.0454) | (0.1325) | (0.0362) | (0.1270) | (0.045) |
| Access to extension | 0.039 | 0.0142 | 0.0343 | 0.0093 | 0.2349 * | 0.083 * |
| | (0.139) | (0.0506) | (0.1568) | (0.0428) | (0.1404) | (0.049) |
| Infrastructure loss | −0.3761 *** | 0.1361 *** | 0.4315 *** | −0.1178 *** | −0.4020 *** | −0.142 *** |
| | (0.086) | (0.0312) | (0.0960) | (0.0260) | (0.0870) | (0.0307) |
| Own processing equipment | −0.274 | −0.0992 | −0.9976 *** | −0.2724 *** | −0.4871 ** | −0.172 |
| | (0.193) | (0.0698) | (0.2870) | (0.0776) | (0.2169) | (0.076) |
| Age of HH | −0.0013 | −0.0004 | −0.0024 | −0.0006 | −0.0031 | −0.001 |
| | (0.0028) | (0.0010) | (0.0031) | (0.0008) | (0.0028) | (0.001) |
| Education HH | −0.3217 *** | −0.1165 *** | −0.2024 *** | −0.0552 *** | −0.2733 *** | −0.096 *** |
| | (0.069) | (0.025) | (0.0764) | (0.0209) | (0.0701) | (0.024) |
| Access to market information | −0.018 | −0.0065 | −0.1943 ** | −0.0530 ** | 0.0110 | 0.0038 |
| | (0.072) | (0.0263) | (0.0774) | (0.0211) | (0.0733) | (0.025) |
| HH involving non-farm activities | 0.400 *** | 0.145 *** | 0.257 ** | 0.070 ** | 0.401 *** | 0.142 *** |
| | (0.102) | (0.036) | (0.112) | (0.031) | (0.104) | (0.036) |
| Constant | −0.249 | | −0.661 | | −0.301 | |
| | (0.163) | | (0.179) | | (0.164) | |

Source: Authors' compilation from the Fadama-III survey (2018). Note: numbers in parentheses are standard errors. *** $p < 0.01$, ** $p < 0.05$, * $p < 0.1$. Note: [†] marginal effects (dy/dx) are evaluated at the sample values and then averaged. HH = Household.

The estimated coefficients from the probit model and their marginal effects reported in Table 5 show that conflict-induced migration was associated with increased severity of household food insecurity measured in all three indicators. Other direct effects of conflicts that were associated with the three food insecurity indicators were household members being traumatized, killed (fatality), loss of property, and injury. We observed from our marginal estimates that trauma due to conflict increases the likelihood of a household experiencing *'ran out of food'* and *'hungry but did not'* over the period under consideration. Similarly, households that lost any member of their household to conflict were more susceptible to experiencing all three food insecurity indicators.

Unlike the case where receiving humanitarian assistance was statistically insignificant in improving the dietary diversity of households, the association between the three food insecurity measures and humanitarian support was negatively related to implying that households that received humanitarian support were better off in terms of the food insecurity experienced the households faced. The implication is that while humanitarian support might not have impacted the dietary diversity of households, it might have, on the other hand, provided short-term succor to ensure households have food to eat at the barest minimum.

## 6. Conclusions and Policy Suggestions

This paper provides two key aspects of conflicts and food security in Nigeria. First, we reviewed the nature and spatial extent of conflicts and general insecurity threats, the drivers, and the implications of conflicts on livelihoods, agricultural production, and food security. We found that the underlying causes for the majority of violent conflicts were competition or access to productive resources, economic inequality, and ethnoreligious tensions. In terms of the types of conflicts, herder–farmer conflicts, Boko Haram insurgency, armed banditry attacks, and communal conflicts are the most widely reported types of conflicts in Nigeria. Our review results further show a spatial variability in the nature and severity of violent conflicts in Nigeria. While Boko Haram insurgency is more prominent in the North-East (spilling to North-West, too), the North-Central is mainly exposed to herder–farmer conflicts, with communal conflicts mostly prominent in the South-South region of the country.

The empirical findings in the paper contribute to the literature on food security effects of conflict-induced shocks in several ways. Firstly, using indicators of experienced-based food insecurity and household dietary diversity indicators, our paper contributes to relevant literature by providing evidence on several pathways through which conflicts affect household food insecurity, which complements existing country-specific evidence. Secondly, we used an additional data set obtained by a rapid interview of households via phone during the COVID-19 major health crisis to explore the confounding implications of conflicts on food security during exogenous external crises, such as COVID-19. We found that conflict-induced shocks significantly reduce household dietary diversity and exacerbate the severity of food insecurity.

Based on the review and empirical findings, we forward the following key policy considerations. (1) The types, causes, and motivations of conflicts appear to differ across states and regions in Nigeria. Therefore, state-level, or region-specific approaches and policy interventions are recommended in addressing triggers of conflicts and their attendant impacts. (2) Given the prevalence of herder–farmer conflicts and the resource use competition as a major driver and the huge collateral impacts that come with it, we suggest policy interventions on property rights and the promotion of an alternative livestock production/management system, such as the practice of cattle ranching. (3) Mitigating conflict-induced shocks, such as forced migrations and fatality, may have significant implications for enhancing household food security in Nigeria. Generally, policies and programs need to be developed to mitigate the direct impact of conflict-induced shocks on households, as they create long-term imbalances that often affect household welfare, including food security.

Finally, we would like to highlight key gaps in conflict research in Nigeria. Most data and available literature often focus on the Boko Haram insurgency, which is the prominent conflict in the North-East, spilling into the North-West, with less attention to other forms of conflicts in Nigeria. Similarly, many studies focus more on the direct effects of conflicts on human life and short-term economic effects and not so much on issues such as gender implications, resilience capacity, forced migration/displaced people, coping strategies, and long-term developmental effects. Future research should aim to unlock various types of conflicts and how they operate at the micro, meso, and macro levels to better understand the nature and pathways to impact food security and the welfare of households.

Finally, we would like to highlight some limitations of the present paper and areas of future research to strengthen the knowledge base on the impact of conflicts on various livelihoods and food security outcomes. First, the results reported in the study are based on a literature survey and basic analytical methods. So, we do not claim the findings as rigorous impact evaluations; instead, the results should be interpreted as the statistical associations between conflict-induced shocks and food security. Secondly, the study is based on limited available data. The World Bank Fadama III (AF-II) data cover only six states out of the thirty-six states of Nigeria. Thus, the study is limited in geographical coverage. Future research could explore the impacts of conflicts on several livelihoods and food security indicators with more comprehensive datasets and rigorous analytical approaches.

**Author Contributions:** Both O.O. and B.B.B. contributed to conceptualization, methodology, formal analysis, writing—original draft, and writing—review and editing. All authors have read and agreed to the published version of the manuscript.

**Funding:** The research output presented here was supported by the Feed the Future Nigeria Agricultural Policy Activity (NAPA) which is funded by the United States Agency for International Development (USAID).

**Institutional Review Board Statement:** Not applicable.

**Informed Consent Statement:** Not applicable.

**Data Availability Statement:** The data used in this study are widely available since it is part of available databases in the public domain. Please see the data section of the paper regarding specific details for data sources.

**Acknowledgments:** We would like to thank the IFPRI-Nigeria research staff for the helpful comments and suggestions provided when the draft paper was presented at the staff seminar held in Abuja. We would also like to express our thanks to three anonymous reviewers whose comments and suggestions helped significantly improve the paper. Finally, our heart appreciation goes to Clare Clingain for her kind assistance in English language editorial support within a day of our request. Any remaining errors are our own.

**Conflicts of Interest:** The authors declare no conflict of interest.

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
