# Peer review of "Conflict-Induced Shocks and Household Food Security in Nigeria"

_sustainability, doi:10.3390/su15065057_

Round 1
Reviewer 1 Report
The topic approach in the paper is interesting. Food security insurance is a problem worldwide. The perspective presented is a current one.
The authors could consider the following recommendations:
· Revise the phrases constructions. In some cases may miss some words (ex. In the we reviewed and synthesized the nature, spatial extent, and implications of conflicts on food security in Nigeria-ln. 8-9). In other, the expressions are too long, and the topic needs improvement.
· Figure 1 resolution needs to be improved.
Yes/No possible answers to questions are quite limiting, restricting the respondent's answering area. In this possible survey, there are a few cases.
The Abstract is an induced idea that will be underlined the major influence of conflictual situations on women. The statistical data analyzed do not evidence such facts.
The manuscript uses two reference systems: the author's name and numerical.
The material is considered an article, but the reference to review type appears more than once in the Abstract and Conclusion parts.
The paper has serious deficiencies in structuralization. A major reconfiguration is recommended.
Weak Reject
Author Response
Detailed responses are uploaded separately as part of the re-submission.

Reviewer 2 Report
Reviewer’s Comments
This study used empirical data and econometric models to investigate the effects of conflicts-induced shocks such as forced migration and fatality on household food security indicators. The presentation of the paper is generally good, except a few mistakes in spelling and grammar. But the major concern is that the authors should use Structural Equation Modeling to integrate all variable as a whole to describe their relationships and contributions to dietary diversity of households at the end of the MS. The authors should address the concerns in the revised MS.
Abstract
Line 8: in the what?
Line 12: our review shows that...
Line 16: please verify whether the South-South region is correct. There is a repeat of two directions.
Keywords
The words should not use the repeated ones appearing in abstract. The authors should use different but important keywords that merely appeared in the title and abstract.
1. Introduction
Line 34: the format of citation is incorrect. Don’t use the format like Kimenyi et al., 2014. In stead, use sequential numbers. Please revised them throughout the paper.
Line 54: in ‘’Aside from the above two major types of conflict,’’, because the authors have used aside from at the beginning of the paragraph. To avoid repetition, I suggest using ‘In addition to’ instead.
Line 60: use based on instead of building on.
Line 70-72: rewrite the following sentence:”Furthermore, evidence on the relationship between household food security, and the direct effects conflict-induced shocks such as forced migration, fatality, loss of property, and injury amongst others, as well as the role these factors play in different pathways to food insecurity.” it is hard to figure out what it means.
Line 75: there is a mistake in ‘an understanding in s highly critical’.
Line 77: are you sure reminder in ‘The reminder of the paper is structure as follows’ is correct. It seems replacing with ‘remaining parts of the paper are organized as follows’ is acceptable.
Line 78: delete extra space.
Line 80: what data are included in the section? Please specify them.
2. Conceptualizing conflicts in Nigeria – drivers and impacts
Line 86: check whether ‘underline on’ is correctly used.
Line 86: the reference of Ningxin (2018) was not given in the reference list.
Line 91: use ‘the ’ before ‘same’.
Line 92: delete the extra space.
Line 101: it seems cum is not a usually used word. Use The suspicion and conflicts, instead.
Line 136:delete the extra space.
3. Overview of the effects of conflicts on food security
Line 159: use correct font for ‘Uppsala Conflict Data’
Line 183:delete the extra space.
Line 200:delete the extra space.
4. Effects of conflicts on food security – empirical analysis
Line 222: use correct font for ‘Detailed description of the phone survey was reported in Balana et al. (2022).’
Line 255: give the equation of Probit model. Simply introduce the model.
Line 262: in equation (1), u is not corrected expressed. Useμin the format instead of u.
Line 289: remove extra dots in Table 2 after the values
Line 313: check the expression p = 0.000.
Line 339: use ‘is consistent with’ in stead of ‘is consistent similar’
Line 376: check the expression: ‘ ...in terms easing of ...‘
5. Conclusions and policy implications
I suggest using Structural Equation Modeling to integrate all variable as a whole to describe their relationships and contributions to dietary diversity of households.
Author Response
Detailed responses are uploaded in a word file.

Reviewer 3 Report
At first glance, it is difficult to see whether the article is a review publication or presents the results and analysis of own research. Most of the article presents conclusions drawn from other publications. However, the authors refer to the results of statistical research (line 216) from the "Fadama III.." project and supplementary research carried out in 2022. The publication would certainly gain value if a subsection "Matrials and methods" describing the methodology of the research was separated the results cited by the authors. Did they participate in the implementation of these studies, was the research methodology identical? Why were only 1,658 (line 225) analyzed out of a total of 1,800 households? Among the questions asked to the respondents were about "about hungry but not eat" or "without eating whole day". How should we understand the respondents' lack of money for food and the simultaneous payment of access to mobile telephony (if the survey was conducted over the phone)? It would certainly be worth mentioning the impact of the war in Ukraine on the reduction of grain supplies to Nigeria in 2022. A chapter in a scientific work should rather not begin with " Figure 1 exhibits.." (line 84). As the statement "about 40 to 70 percent of household survey..." (lines 296-297) should be understood, it is a very wide range for the results of precise statistical surveys... In "Abstract" (line 8) the sentence seems to be incomplete. On what basis (research) do the authors write about "gender dimensions.." (line 385)? The attached "References" lack any order, which makes them very difficult to review.
Author Response
The file containing our detailed responses is uploaded.

Round 2
Reviewer 1 Report
The authors have improved some aspects. The main issues remain undiscussed.
1) The principal unsolved aspect is the English style.
2) Remains also the aspect related to the limitation questions that Authors considered initially.
Weak Accept
Author Response
Responses file attached.

Round 3
Reviewer 1 Report
The authors considered the recommendations made.